# Exosomes: Insights from Retinoblastoma and Other Eye Cancers

**DOI:** 10.3390/ijms21197055

**Published:** 2020-09-25

**Authors:** Kashmiri Lande, Jitesh Gupta, Ravi Ranjan, Manjari Kiran, Luis Fernando Torres Solis, Arturo Solís Herrera, Gjumrakch Aliev, Roy Karnati

**Affiliations:** 1Department of Animal Biology, University of Hyderabad, Hyderabad 500046, India; kashmirilande0412@gmail.com (K.L.); jiteshgupta2013@gmail.com (J.G.); anandranjanravi16@gmail.com (R.R.); 2Department of Systems and Computational Biology, University of Hyderabad, Hyderabad 500046, India; manjari.hcu@uohyd.ac.in; 3The School of Medicine, Universidad Autónoma de Aguascalientes, Aguascalientes 20392, Mexico; lfts99@yahoo.com.mx; 4Human Photosynthesis© Research Centre, Aguascalientes 20000, Mexico; comagua2000@gmail.com; 5I.M. Sechenov First Moscow State Medical University (Sechenov University), St. Trubetskaya, 8, bld. 2, 119991 Moscow, Russia; 6Research Institute of Human Morphology, Russian Academy of Medical Science, Street Tsyurupa 3, 117418 Moscow, Russia; 7Institute of Physiologically Active Compounds, Russian Academy of Sciences, Chernogolovka, 142432 Moscow, Russia; 8GALLY International Research Institute, 7733 Louis Pasteur Drive, #330, San Antonio, TX 78229, USA

**Keywords:** exosomes, miRNA, retinoblastoma, tears, cornea, uveal melanoma

## Abstract

Exosomes, considered as cell debris or garbage bags, have been later characterized as nanometer-sized extracellular double-membrane lipid bilayer bio-vesicles secreted by the fusion of vesicular bodies with the plasma membrane. The constituents and the rate of exosomes formation differ in different pathophysiological conditions. Exosomes are also observed and studied in different parts of the eye, like the retina, cornea, aqueous, and vitreous humor. Tear fluid consists of exosomes that are shown to regulate various cellular processes. The role of exosomes in eye cancers, especially retinoblastoma (RB), is not well explored, although few studies point towards their presence. Retinoblastoma is an intraocular tumor that constitutes 3% of cases of cancer in children. Diagnosis of RB may require invasive procedures, which might lead to the spread of the disease to other parts. Due to this reason, better ways of diagnosis are being explored. Studies on the exosomes in RB tumors and serum might help designing better diagnostic approaches for RB. In this article, we reviewed studies on exosomes in the eye, with a special emphasis on RB. We also reviewed miRNAs expressed in RB tumor, serum, and cell lines and analyzed the targets of these miRNAs from the proteins identified in the RB tumor exosomes. hsa-miR-494 and hsa-miR-9, upregulated and downregulated, respectively in RB, have the maximum number of targets. Although oppositely regulated, they share the same targets in the proteins identified in RB tumor exosomes. Overall this review provides the up-to-date progress in the area of eye exosome research, with an emphasis on RB.

## 1. Introduction: Exosomes

Exosomes were considered as waste bags or the organelle responsible for the storage of cell debris but later characterized in 1987 [1] as nanometer-sized extracellular double-membrane lipid bilayer bio-vesicles secreted once multivesicular bodies gets fused with the plasma membrane [2]. These contain an entire cargo of protein, DNA, RNA (tRNA, mRNA, miRNA, lncRNAs), lipids, and sugars [2,3,4,5]. They are heterogeneous and dynamic in nature, constituting markers, or contents of cells that secrete them [4]. All eukaryotic cells are known to secrete exosomes that are detected in plasma and various fluids, such as amniotic fluid, bile, breast milk, cerebrospinal fluid (CSF), saliva, semen, synovial fluid, tears, and urine. The secretion rate changes during certain pathological conditions, which can be used for early detection of disease conditions [2,4,5,6,7,8]. Exosomes are also detected in the growth medium of all cell cultures. The constituents of exosomes are responsible for functions, such as activation of cell signaling pathways, cell-cell communication, modulation of immunity, cargo shuttle from donor to the target cell, and antigen presentation [2,4,5,9]. Exosomal biogenesis requires the invagination of late endosomes within multivesicular bodies (MVBs), resulting in the formation of intraluminal vesicles (ILVs) [10]. ESCRT (endosomal sorting complex required for transport) is critical in the formation of ILVs and is required for MVB formation, vesicular budding, protein cargo sorting [11,12]. Tetraspanin-associated microdomains (TEMs) play a role in exosome formation and protein cargo sorting [13,14,15]. Recently, the ceramide pathway, which is ESCRT independent, has been recognized to transport protein cargo where lipids seem to play a major role in exosomes biogenesis [16]. A total of 11,261 proteins, 2375 mRNAs, and 764 miRNAs, listed in the Exocarta database, are identified from 134 exosomal studies [17]. Exosomes are enriched in proteins like tetraspanins (cell fusion and invasion proteins) and heat shock proteins (antigen binding and presentation); Tumor Susceptibility Gene (TSG)101 (MVB formation proteins, exosome biogenesis), membrane transport, and fusion serve as potential exosome markers. Exosomes constituents also include various lipids, such as sphingomyelin, cholesterol, phosphatidylserine, glycosphingolipids, which play a structural and functional role in their formation and release in the extracellular environment [18]. As genetic material can be transported, exosomes are responsible for altering the genetic constitution of the target cell. The ubiquitous presence of exosomes and the emerging research of finding specific constituents of exosomes and their functional characterization has paved the way in the medical field for finding the cause, diagnosis, and therapy involving target specific markers of exosomes.

## 2. Exosomes of the Eye

Exosomes, present in tears, aqueous humor, vitreous humor, are easily accessible and may be explored for studies involving diagnosis, therapy, and drug efficacy (Figure 1). Exosomal functions identified via various studies include intercellular communication, extracellular matrix (ECM) communication with cells, ECM assembly and remodeling, adhesion, and cellular waste removal [19,20]. The eye is an exosome-mediated immune-privileged site similar to the placenta, owing to Fas L and TRAIL ligands and functions in the prevention of inflammation [21,22,23,24]. Exosomes are involved in angiogenesis or neovascularization. Hypoxia in tumor cells leads to the instant release of exosomes, which, when taken up by other cells, stimulate angiogenesis and further metastasis [25,26,27,28]. Exosomes released from tumor cells may also cause leakiness in the cell membrane because of their cellular content. miR105 from exosome is shown to downregulate ZO1, a tight junction protein that increases vascular permeability and helps in metastasis [25,29]. Other studies have shown that exosomal miRNA 146a expression is upregulated in serum as well as vitreous humor, suggesting that it may well be a biomarker for the diagnosis of uveal melanoma (UM) [25,30].

### 2.1. Tear Fluid Exosomes

Exosomes are found in tears in normal as well as cancer patients. Cancer-specific markers are highly expressed in a study conducted in metastatic breast cancer patients [31]. Sjogren’s syndrome (SS) is an autoimmune disease that targets glandular tissues, including the lacrimal glands and exocrine salivary glands, where local inflammation is thought to manifest primary symptoms, such as xerostomia and xerophthalmia. The presence of T cell exosome-derived miR-142-3p in SS suggests it as a potential driver in the immunopathology [32]; also, exosomes are already proven to transfer miRNA between cells [33], and it has a functional role in the glandular cell dysfunction [32]. Breast cancer-specific miR-21 and miR-200c show higher expression levels in tear exosomes of metastatic breast cancer patients as opposed to healthy controls [31]. Most of the studies suggest exosomal constituents of tears to be biomarkers.

### 2.2. Retinal Exosomes

The immune-privileged nature of the eye has provided protection against inflammation and compromised vision [34] as the retina fails to regenerate after severe inflammation. The retinal pigment epithelium (RPE), which is present between the retina and choroid, forms the blood-retina barrier and is one of the key sites for the pathophysiology of AMD (age-related macular degeneration). For visual homeostasis and proper vision, communication between RPE, photoreceptors, and choroid capillary endothelium should be achieved. Exosomes exchange cargo between neighboring cells and have gained importance in recent years. Different cargos are observed in exosomes from RPE cells under stress [35,36,37].

When ROS (reactive oxidation species) production in the RPE increases, more exosomes with VEGF (vascular endothelial growth factor) or VEGF 2 receptors are released, which are responsible for accelerated choroidal neovascularization [36,38,39]. When ROS increases, it leads to the secretion of autophagosomes, lysosomes, and exosomes or MVBs as they combine and work together in degrading the content [40,41,42,43]. Thus, the overproduction of ROS increases stress in cells, which influences the exosomal release and autophagic activity. The exosomes are released as a result of stress-damaged nearby cells, leading to degeneration, which is one of the many reasons for the pathophysiology of AMD [44]. Besides, recent findings suggest the potential role of extracellular vesicles (EVs) and apoptotic blebs in the dysregulation of complement pathway by removing cell surface complement immune regulators, such as CD46, CD59, CD55 from RPE (in ARPE-19 cell model), leaving the RPE cells more vulnerable to complement attacks under oxidative stress [45,46,47]. Here, exosomes’ presence suggests its role in causing trouble owing to oxidative stress.

### 2.3. Corneal Exosomes

The cornea is avascular, maintained in the state owing to the balance between pro-angiogenic and anti-angiogenic stimuli. Any imbalance in stimuli or inflammation in the cornea may lead to neovascularization, causing opacity and vision loss [48,49]. Extracellular vesicles in corneas are responsible for cell-cell communication and have also been shown to interact with corneal keratinocytes/fibroblasts during fibrosis and cornea scarring, suggested by the high expression of SMA (α-smooth muscle actin) [50]. Besides, studies have shown that corneal epithelium-derived exosomes are one of the factors responsible for wound healing and neovascularization by stimulating endothelial cell proliferation [51]. Corneal graft rejection is also one of the main concerns; even though cornea is immune privileged, graft rejection occurs when major histocompatibility complex (MHC) antigens from a donor are recognized by T cells, i.e., adaptive immune response [52]. Studies have held EVs, including exosomes, partly responsible for the same [53]. Whereas other studies have shown that exosomes from certain immuno-suppressive cell populations ensure allograft survival by providing tolerance [54].

### 2.4. Exosomes of Aqueous Humor

Aqueous humor, a watery transparent fluid, is secreted by the ciliary body, which nourishes the cornea and lens and gives shape to the eye by maintaining intraocular pressure. Aqueous humor also consists of exosomes where one such study has identified more than 10 miRNA isolated from aqueous humor exosomes collected during cataract surgery, in which miR-486-5p, miR-204, and miR-184 are abundant [55].

### 2.5. Exosomes of Vitreous Humor

Vitreous humor, a gelatinous transparent fluid between the lens and retina of the eye, is produced by non-pigmented cells of the ciliary body, a part of the Uvea. The UM patients are shown to have different miRNA profiles of vitreous humor and serum with respect to healthy individuals [30].

## 3. Eye Cancers

Eye cancers are rare but are of unique importance as both vision and quality of life are severely compromised. Eye cancer can be characterized based on regions, intraocular cancers (occurs within the eyeball), orbital cancers (includes eye socket), adnexal cancers (including accessory regions, such as eyelids and tear glands). Intraocular cancers are of two types; primary, which starts primarily in the eyeball, of which most common is RB, and secondary, metastasized to eyeball from different parts of the body, mostly from breast and lung tumors. Common eye cancers include RB in children and UM occurring in the choroid.

Uveal melanoma collectively comprises melanoma of iris, choroid, and ciliary body, which are uncontrollable divisions of melanocytes residing in the uveal tract [56]. They are mostly seen in older people, and the risk increases with age [57,58]. The occurrence in children and the congenital cause are very rare [59,60]. With respect to gender, the incidence is more prevalent in males than females in higher than 65 years of age and has a higher occurrence in the white population than the black population [57,61,62,63,64]. However, the larger population studies show equal prevalence in the male and female populations [65]. The occurrence of tumors in UM is in three sites: iris (3–4%), ciliary body (6–7%), choroid (90%) [66]. Iris melanoma is often spotted early [65]. Microscopy studies have shown that 5% of the sample population shows metastasis, and older age combined with intraocular pressure and extraocular extension increases the risk of metastasis [67]. Diffuse iris melanoma is a rare form of iris melanoma, and glaucoma and cataract are the main complications. Choroidal and ciliary body melanomas have symptoms, such as photopsia, visual field loss, visible tumor, blur vision, pain, metamorphopsia, different combinations of symptoms, and also asymptomatic [68]. In addition to this, ciliary body melanoma often hides behind iris and, thus, requires more prominent symptoms for prognosis. The pathogenesis for UM involves genetic and molecular changes that include disruption in the Rb (retinoblastoma) tumor suppressor pathway, where cyclin D overexpression directly and the methylation and inactivation of the INK4A gene indirectly inactivate Rb by hyperphosphorylation [68,69,70]. The diagnosis of UM varies for iris, choroid, and ciliary body, also depending on the symptoms and rate of progression, mostly including CT (computed tomography), high-frequency ultrasound, optical coherence tomography, MRI (magnetic resonance imaging), and invasive biopsy. The most common treatments for local UM include laser therapy, enucleation, resection through surgery, brachytherapy, and radiation. Treatments after metastasis include chemotherapy and immunotherapy [71]. The alterations in the expression of miRNA are common in the vitreous humor and vitreal exosomes, where miR-146, detected in serum, serum exosomes, and tissue, could serve as a potential marker of UM [30]. Exosomes isolated from the liver perfusion are of melanoma origin in metastatic UM patients, and the constituents of these exosomes are different from the healthy controls [72]. These might serve as markers to monitor prognosis and disease progression [73]. The majority of the proteins secreted in UM is through exosomal mechanisms, and they are involved in various cellular mechanisms like extracellular matrix remodeling and metastasis [74].

Tumors of conjunctiva and cornea, basal cell carcinoma of the eyelid, squamous cell carcinoma, hemangioma, intraocular lymphomas, and lacrimal gland tumors are other types of eye cancers. Exosomes derived from the corneal epithelium, corneal keratinocytes, and stroma are involved in cell-cell communication, migration, cell proliferation, and wound healing [75,76,77]. Pterygium, a surface lesion from conjunctiva towards the cornea, is due to the proliferation and ECM remodeling. The secreted proteins from the pterygium are of exosome origin [78]. However, to date, no studies have been reported on exosomes and their role in corneal and conjunctival cancers. Tumors of conjunctiva and cornea include tumors from benign lesions to malignant Kaposi’s sarcoma. A neoplasm in these segments can arise from epithelial as well as stromal structure, the former being more common. Some of them are congenitally present or arisen after birth [79]. They are rare but can spread through blood and have proven life-threatening. Kaposi’s sarcoma-associated herpesvirus (KSHV) is shown to change the microenvironment by utilizing the host-derived exosomes [80]. Epidermal growth factor receptor plays an important role in the infection of KSHV, and this is promoted by the exosomes associated with HIV [81].

Exosomes and their role in basal cell carcinoma of eyelids, hemangioma, and squamous cell carcinoma of the eye are not well studied. Basal cell carcinoma is the most common cancer of eyelids. Metastasis is rare, but if they do, they can affect various regions of the brain. Therefore, various surgeries, radiation, and chemotherapy are suggested. Squamous cell carcinoma can metastasize at a much faster rate and prove fatal if the prognosis is late. Hemangioma is a benign tumor of the retina and choroid, which, from its name, suggests the uncontrolled proliferation of blood vessels. It may sometimes lead to bulging of the eye. Intraocular lymphomas (IOL) can be primary or secondary. They can arise in the uvea, optic nerve, vitreous, or can occur somewhere outside and metastasize to the eye. The origin is from B cells, but there are T cell variants too. It is a rare type of malignancy. Lacrimal gland tumors range from benign epithelial and lymphoid lesions to aggressive malignant carcinomas and sarcomas. Surgery is used to treat benign tumors, whereas adjuvant therapy or chemotherapy and other high invasive procedures are required to treat malignant ones. The research regarding eye cancer has led to more knowledge in the overall cancer biology, cell cycle, development, cell differentiation, and diverse diagnostic and treatment strategies. Exosomes intercellular communications in tumorigenesis and metastasis are major players in the progression of cancer. Besides, the extracellular matrix-modulating exosomal activity by exosomal metalloproteinases, syndecans (proteoglycans), annexins has been identified as constituents responsible for the progression of metastasis [82,83,84].

## 4. Retinoblastoma and Exosomes

Retinoblastoma is a primary intraocular and the most common form of cancer in children [85]. It occurs in two types: (1) Bilateral/multifocal (in both eyes), which is heritable, occurs because of germline mutation of the RB1 tumor suppressor gene in a developing retinal cell, and includes 25% of the total cases, and (2) Unilateral/unifocal (in one eye), which is non-heritable and includes 75% of all cases [86]. Tumor formation is because of the loss of function mutation in both the alleles of the RB1 gene present in chromosome 13q [87,88]. The RB1 gene codes for tumor suppressor retinoblastoma protein that binds to the transcription factors and proteins, which remodel chromatin, regulating the cellular replication [88,89,90]. The disease staging is done by the international RB staging system (IRSS). RB seeding may also occur in advanced stages when the tumor disperses into a liquid or semi-liquid compartment. Intraocular RB can seed to four sites: (1) Vitreous seeding—by endophytic disruption of the internal limiting membrane (ILM) of the retina at the tumor apex, (2) Retrohyaloid space—by endophytic disruption of ILM at tumor base, (3) Subretinal space by exophytic growth, and (4) Aqueous fluid of anterior and posterior chamber [91]. Seeding may lead to dust, cloud, or sphere formation.

The diagnosis for RB is tricky as it may be the reason for spreading the tumor outside the eye (extraocular). Besides, most of the time, in the starting stage, tumor cells are not found in the blood of the patients. So, recently, researchers are focused on finding alternative ways for diagnosis, which are less or non-invasive and will not spread the tumor. The exosomes are abundantly secreted from aggressive tumor cells and are present in various biological fluids [92]; liquid biopsies may promise to provide an alternative diagnosis for RB. Besides, proteins and non-coding RNAs, specifically miRNAs of exosomes, are identified, isolated, and characterized for their role in various pathological conditions [93,94]. In the recent study, the proteins of exosomes are characterized by the RB tumor tissue and from the RB vitreous seeding (RBVS) in the vitreous humor [95]. In RBVS exosomes, the proteins involved in the invasion and metastasis are identified, regulating functions, such as extracellular matrix (ECM) remodeling and metabolism of glucose and amino acids, etc. [95]. Besides, to prognosticate RB, less invasive procedures, such as a liquid biopsy, would serve as the best option for children and are less painful than the invasive procedures, such as bone marrow aspiration or lumbar puncture, for the detection of metastasis. Therefore, circulating exosomal biomarkers are considered one of the best choices for the diagnosis as well as efficacy.

## 5. miRNA and Their Targets in Retinoblastoma

miRNAs are non-coding RNAs that have a regulatory role. Circulating miRNAs have been reported in serum as potential biomarkers for various cancers [96,97]. Exosomal miRNAs are shown as potential biomarkers for the diagnosis of various cancers [98,99,100,101,102,103,104]. A cluster of miRNA, including hsa-miR-494, is identified to be highly expressed in human RB tissues and maybe a major constituent involved in tumorigenesis [105]. In another bioinformatic analysis, 22 miRNAs are upregulated in SNOUT-RB1 cell lines, including miR-29c, and other 17 are upregulated in Y79 cell line, including hsa-let-7i, and are found to be related to biological processes and could affect processes in cell cycle and cell adhesion, suggesting a role in the treatment of RB by targeting miRNA [106]. Bioinformatics analysis and next-generation sequencing have confirmed the relative expression pattern of hsa-let-7i, suggesting its role in the pathogenesis of myopia [107]. miR-9 is identified along with other miRNAs as circulating small RNAs in the vitreous humor of patients with ocular diseases but is not detectable in serum [108]. Another study has shown the suppression of UM by miR-9 through the NFκB1 pathway via stopping cell migration and invasion [109]. The higher expression of miR-148a in the ocular region is detected in patients with retinal detachment (RD), suggesting that it might function in EMT (epithelial-mesenchymal transition) in retinal pigment epithelium [110]. The correlation of highly expressed miR-148a in a vitreous fluid is shown with the duration and severity of the rhegmatogenous retinal detachment disease [111].

The studies by Beta M et al. and Ravishankar H et al. have identified the miRNA constituents of exosomes in the RB cell lines, WERI-Rb-1, and NCC-RbC-51 cells and in RB tumor samples and RB patients’ serum samples. hsa-miR-1307-5p, hsa-miR-301b-3p, hsa-miR-148a-5p, and hsa-miR-216b-5p are present in exosomes of both cell lines, whereas hsa-miR-582-3p and hsa-miR-887-3p are specific to WERI and hsa-miR-200a-5p and hsa-miR-483-5p to NCC-RbC-51 cells. A total of 33 miRNAs are common in the miRNAs deregulated in the RB tumor and those present in the serum of RB children. Of these, 25 are upregulated, and eight are downregulated [112,113]. In another study by Galardi A et al., the proteomic profiling of the exosomes constituents has identified 99 proteins that are exclusively present in RB tumors [114]. We have analyzed the targets of the miRNAs from both these studies [112,113] from the proteins identified in RB tumor exosomes [114]. hsa-miR-494 (upregulated) and hsa-miR-9 (downregulated) have the maximum number of targets in proteins identified in RB tumor exosomes (Table 1).

The miRNAs common to RB tumor and RB serum and those present in RB cell line exosomes are shown to function as important regulators in various studies; hsa-miR-216b-5p and hsa-miR-301b-3p are upregulated in the primary RB tissues and RB cell lines [112]. hsa-miR-30b is one of the 25 upregulated miRNAs in the comparison of miRNA between primary RB tissues and serum obtained from children suffering from RB [113].

In the upregulated miRNAs of RB, miR-142-3p is considered as a probable target for the treatment of UM as its overexpression has shown to suppress UM, inhibiting cell migration, proliferation, and invasiveness by targeting CDC25C, TGFβR1, GNAQ, WASL, and RAC1 [115]. Another study has revealed that the overexpression of miR-142 causes apoptosis-associated protein expression in osteosarcoma cells, suggesting the possible role of miRNA in the osteosarcoma treatment [115]. Runx3, a tumor suppressor, is upregulated after miR-106b suppression, and the inhibition of Y79 cell proliferation is seen, which suggests that Runx3 is a target of miR-106b in an RB [116]. miR-182 alteration is related to cervical cancer pathogenesis and plays an oncogenic role involving apoptosis and cell cycle pathways [117]. miR-182 as an integrator of growth, apoptosis, differentiation, tumor progression in glioblastoma [118]. miR-182 and miR-183 are responsible for promoting cell proliferation and invasion in mesothelioma by specifically targeting FOXO1 and p27 [119]. miR-148b targets the WNT1/β catenin pathway and functions as a suppressor of cell proliferation and invasion in hepatocellular carcinoma (HCC), suggesting that it may be a possible target of HCC and may prove as a treatment strategy [120]. miR-148 is shown to regulate Mitf expression in melanocytes, and loss of this particular regulation may be a particular factor leading to the formation or progression of melanoma [121]. miR-29c is shown to target LINC01296, an oncogene that provides a therapeutic option in ovarian cancer [122]. miR-30b-5p plays a role in the suppression of the tumor in esophageal squamous cell carcinoma [123]. miR-494 promotes cancer proliferation in breast cancer cell lines and colorectal cancer [124,125]. But in another study, it is shown to inhibit cancer proliferation in breast cancer via the inhibition of PAK1 [126].

In the miRNAs that are downregulated in RB, miR-let-7a, a tumor suppressor, inhibits cellular proliferation and suppresses tumor growth by inhibiting E2F2 in osteosarcoma cells [127]. miR-92a, a tumor suppressor, regulates primary mediastinal large B-cell lymphoma (PMBL) by targeting FOXP1, known to deliver an oncogenic effect [128]. miR-216a plays a role in the inhibition of gastric cancer metastasis via targeting JAK2/STAT3-mediated EMT, thereby suggesting its role in tumor suppression and, ultimately, gastric cancer development [129]. hsa-miR-9 plays a role in the prognosis of glioblastoma and is shown to control metastasis by regulating MAPK4 signaling [130]. miR-9 downregulates TM4SF1 and is shown to function for the suppression of metastasis in colorectal cancer [131].

Among the miRNA studied in cell lines, miR148a targets CCK-BR via the inactivation of STAT3 and Akt and is identified as a tumor suppressor in human gastric cancer [132]. A recent study also suggests the role of miR-148 as a suppressor of hepatocellular carcinoma tumor where sphingosine-1-phosphate receptor 1 is the target [133]. A study has reported that hsa-miR-582-5p, an oncogenic miRNA, and miRNA-363 are responsible for human glioblastoma cell survival by effectively targeting caspase 3, caspase 9, and Bim [134]. Another study has revealed the role of miR-582-5p as a tumor-suppressor in gastric cancer cell growth via targeting AKT3, suggesting it as a target for treatment as well as diagnosis of gastric cancer [135]. hsa-miR-582, along with hsa-miR-320d, is responsible for the regulation of apoptotic activity in vascular smooth muscle cells [136]. Upregulated miR-200a provides resistance to the treatment for chemotherapy by antagonizing TP53INP1 and YAP1 in breast cancer cell lines [137]. miR-200a is reported to confer tumor-suppressive ability by regulating Wnt/β-catenin signaling pathway and is considered to be a potential candidate for treatment [138]. Decreased expression of miR-200 family members has shown resistance against the endocrine antiestrogen in LY2 human breast cancer cells [139]. miR-483-5p, in coordination with miR-125-3p, is identified as a promoter of adipogenesis via the suppression of the RhoA/ROCK1/ERK1/2 pathway, and their studies may prove as a strategy to treat obesity or multiple symmetric lipomatosis (MSL) [140]. miR-483 is responsible for playing a role in digestive tract cancers (DCT) [141]. miR-216b is shown to be responsible for the tumor suppression of pancreatic ductal adenocarcinoma cells (PDAC) and is suggested as a treatment target in PDAC [142]. miR216b targets FOXM1 and inhibits cell proliferation in cervical cancer [143]. A study in prostate cancer patients has revealed the possible oncogenic role of the miR-130b/301b cluster [144].

Interestingly, in the proteins identified in the exosomes of RB tumor, 23 proteins are common targets to both hsa-miR-494, which is upregulated, and hsa-miR-9, which is downregulated, in RB tumor and RB serum (Table 2). This even shows the complex regulation of miRNA in the tumorigenesis of RB. The proteins identified in the exosomes of RB tumor are shown to have a role in cell growth and tumorigenesis, as discussed. Proteins abl interactor 2 (ABI2), neurocan (NCAN), V-type proton ATPase subunit C1 (ATP6V1C1), and UBX domain protein 4 (UBXN4) are identified in RBT-derived exosomes [114]. Studies have shown that ABI2 (Abl-interactor 2 protein) suppresses cell growth, and its truncated form plays a role in the acceleration of tumorigenesis, suggesting that ABI2 may be a tumor suppressor in RB [145,146]. NCAN, an extracellular proteoglycan, is shown to be responsible for promoting malignancy by the stimulation of neuroblastoma cells [147].

ARPP19 (cAMP-regulated phosphoprotein 19) promotes mitotic entry by inhibiting PP2A [148]. In addition to miR494 and miR 9, ARPP19 is shown to be a target of miR 320 a and miR 26 a in breast [149] and thyroid cancers [150]. ATPase plasma membrane Ca2+ transporting 2 (ATP2B2) is a calcium transporter, whose levels correlate with mortality in breast cancers [151]. The overexpression of ATP6V1C1 is seen in a variety of cancers, including breast cancer, oral cancer, and oral squamous cell carcinoma [152,153,154]. ATP6V1C1 controls tumor growth and bone metastasis, and the silencing of the gene suggests treatment and prevention strategies against cancer [154]. The overexpression of atp6v1c1 facilitates filament actin arrangement in the metastasis of cancerous cells and proves an innovative target in the treatment of breast cancer metastasis [153]. The overexpression of camsap1 mRNA is seen in laryngeal squamous cell carcinoma rather than normal tissues [155]. Spectrin-associated protein 1 (Camsap1) is shown to be modulated by miR 26; this modulation alters the tumor microenvironment and inhibits metastasis [155]. The expression of cathepsin V (CTSV) and human augmin complex unit 3 (Haus3) correlates with the poor prognosis in invasive and ductal carcinoma in situ types of breast cancer [156] and hepatocellular carcinoma, respectively [157]. CISD2 (CDGSH iron-sulfur domain (2)) promotes glioma cell proliferation via the inhibition of beclin-1-mediated autophagy [158]. CUX1 (CUT-like homeobox (1)) is shown to be related to both progression and suppression of tumor, but a haploinsufficient tumor suppressor gene and its overexpression are seen in advanced cancers [159]. DCC (deleted in colorectal carcinoma), a tumor suppressor gene, whose functional loss in colorectal cancer suggests its metastatic role [160]. Another study has suggested its role in tumor cell differentiation and tumor proliferation [161]. The downregulation of the DPF3 (double PHD fingers) gene activates the JAK2/STAT3 signaling pathway and performs a major role in the progression of breast cancer [162]. KDM1B (flavin-dependent histone demethylases) knockdown shows to inhibit cellular proliferation and induce apoptotic activity in pancreatic cancer and suggests a role in prevention [163]. Intersectin 1 (ITSN1) has the transforming potential and is shown to be involved in the tumorigenesis of neuroblastoma [164]. MTMR2 (myotubularin-related protein-2) is shown to inactivate IFN γ/STAT signaling and promote gastric cancer (GC) invasion and metastasis and maybe a new therapeutic target for the treatment of GC [165]. Unphosphorylated PEA-15 (astrocytic phosphoprotein) is responsible for tumor progression by blocking the β-catenin pathway [166].

## 6. Conclusions and Future Perspectives

Extensive research on exosome biogenesis and their role in the pathology of RB and other eye cancers will help in the development of new diagnostic approaches and therapeutic strategies. The diagnosis of RB requiring invasive procedures will benefit immensely if relevant disease-specific exosome or its constituents are discovered in the blood. The subtle differences in the exosomes during therapy will also help in streamlining the drugs and their doses. Unlike other cancers, the research on exosomes in RB and other eye cancers is not well explored. Tear proteome is largely uncharacterized. The focus on identifying the potential biomarkers of eye cancers by studying the exosomes in aqueous humor and vitreous humor and tears will help to save the globe of the eye from invasive procedures, both for diagnosis and therapy.

## Figures and Tables

**Figure 1 ijms-21-07055-f001:**
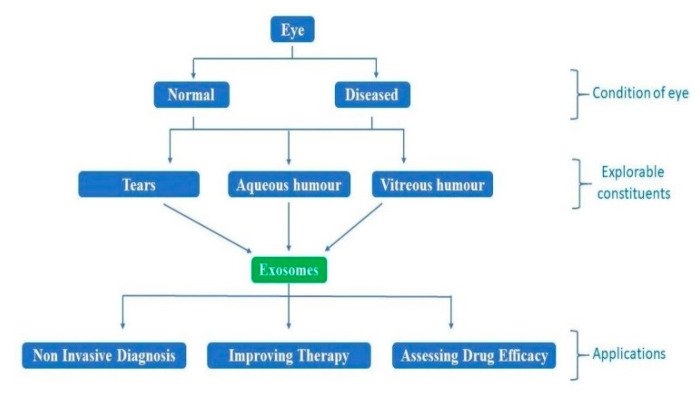
The schematic drawing of how the exosomes may be studied for various applications in the pathophysiology of the retinoblastoma and other eye cancers.

**Table 1 ijms-21-07055-t001:** The list of miRNA reported in RB tumor, RB serum, and RB cell lines and their targets from the proteins identified in the RB tumor exosomes.

Upregulated miRNA RB Tumor and RB Serum and Targets Identified in RB Tumor Exosomes		Downregulated miRNA RB Tumor and RB Serum and Targets Identified in RB Tumor Exosomes		miRNA Derived from RB Cell Lines Exosomes and Targets Identified in RB Tumor Exosomes	
miRNA	Target Gene	miRNA	Target Gene	miRNA	Target Gene
hsa-miR-16-1	KIAA1598	hsa-let-7a	TJP3	hsa-miR-148a	CAMSAP1, IKBIP, NUP50, PPP2R2A
hsa-miR-29b-1	LPGAT1	hsa-miR-92a	INTS1		
hsa-miR-34a	PTRHD1	hsa-let-7f-2	ERCC3, UNC13B		
hsa-miR-96	UNC13B				
hsa-miR-143	ADD3, AK2	hsa-miR-217	ARPC2, EIF1	hsa-miR-582	CACNA2D2, MPP1, NEFL, PIN4, VDAC3
hsa-miR-30d	DCK, POLD3	hsa-let-7a-2	CCNB1, GSTM1, TMEM55B		
hsa-miR-16	OS9, TPPP3	hsa-miR-92a-1	FAM120A, MPDZ, TNR, TPPP3	hsa-miR-200a	CTSV, DAG1, GLRX, MTMR2, PRPS1, SLC27A4, UBXN4
hsa-miR-142	CCNB1, FAM49B, SDHB				
hsa-miR-106b	PPP2R2A, UBXN4, WDR36	hsa-miR-216a	GLRX, POLD3, PPP2R2A, DLG5, PPIF, UNC119		
hsa-miR-182	IKBIP, LMAN2, NUP50				
hsa-miR-183	MPP1, SEC61B, SNAP25				
hsa-miR-148b	MPDZ, TRIM2, DKFZp781M17165, LONP1, PPP1R2	hsa-miR-92a-2	CACNA2D2, MPP1, OS9, P4HTM, PAK1, SIN3B	hsa-miR-483	DCK, DKFZp781M17165, GSTM1, HAUS3, MYDGF, PPP2R1B, PTRHD1
hsa-miR-29c	DAG1, NDUFS6, PDCD4, PPIF, NEFL, PRPS1	hsa-let-7i	DNAJC5, ATP8A1, DAG1, PTRHD1, SLC25A13, TTC9C, UQCRFS1	hsa-miR-216b	ABI2, ATP2B2, ATP8A1, CCDC85C, CISD2, CUX1, DNAJC5, FAM120A, ITSN1, LPGAT1, PPP1R2, RQCD1, TNR, TRMT6, TSPAN14, UBXN7, WIPF2
hsa-miR-30b	CACNA2D2, ERCC3, GSTM1, SLC27A4, TMEM55B, MED4, ARPC2				
hsa-miR-494	ABI2, ATP2B2, ATP6V1C1, ATP8A1, CAMSAP1, CCDC85C, CISD2, CTSV, CUX1, DCC, DLG5, DNAJC5, DPF3, EIF1, FAM120A, GLRX, HAUS3, ITSN1, KDM1B, MTMR2, NCAN, PAK1, QSER1, SLC25A13, UBXN7, WDR11, WIPF2, ARPP19, PEA15, PIN4, PPP2R1B, RQCD1, SIN3B, SYNGR1, TNR, TRMT6, TSPAN14, UNC119, UQCRFS1	hsa-miR-9	ADD3, ARPP19, ATP6V1C1, CCDC85C, CISD2, CTSV, CUX1, DCC, DCK, FAM49B, ITSN1, KDM1B, LPGAT1, MED4, NCAN, PDCD4, PEA15, QSER1, RQCD1, TRIM2, UBXN4, WDR11, ABI2, ATP2B2, CAMSAP1, DKFZp781M17165, DPF3, HAUS3, IKBIP, MTMR2, NEFL, PPP1R2, PRPS1, SLC27A4, SNAP25, TSPAN14, UBXN7, WIPF2	hsa-miR-301b	ADD3ARPP19ATP6V1C1CCNB1DCCDLG5DPF3FAM49BNCANPAK1PEA15QSER1SLC25A13SNAP25TMEM55BTRIM2UQCRFS1WDR11

**Table 2 ijms-21-07055-t002:** Common targets of hsa-miR-494 (upregulated) and hsa-miR-9 (downregulated) in RB tumor and RB serum.

ABI2	CAMSAP1	CUX1	ITSN1	PEA15	UBXN7
ARPP19	CCDC85C	DCC	KDM1B	QSER1	WDR11
ATP2B2	CISD2	DPF3	MTMR2	RQCD1	WIPF2
ATP6V1C1	CTSV	HAUS3	NCAN	TSPAN14

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
