# Peer review of "Exosomes: Insights from Retinoblastoma and Other Eye Cancers"

_ijms, 2020, doi:10.3390/ijms21197055_

Round 1
Reviewer 1 Report
The review by Kashmiri Lande and colleagues summarizes the literature regarding the role of exosomes in retinoblastoma and other eye cancers. The authors reviewed miRNAs expressed in retinoblastoma tumor, serum and cell lines and analyzed the targets of these miRNAs from the proteins identified in the retinoblastoma tumor exosomes.
This review is an interesting overview on the role of miRNAs and specifically exosomes in retinoblastoma. It is well written even if the English language should be revised.
Please check and uniform all the abbreviation in the text: page 1 line 41 RB is abbreviate but not in the rest of the paragraph. The same issue in page 6 line 223 and 233. Page 8 lane 269 uveal melanoma is abbreviate as UM, but not mentioned for the first time.
The figure 1 is not consistent with the sentence reported at page 4, lanes 148 to 154. Perhaps the authors refer to another figure. Please check
Table 1 is unclear. We recommend to add information on the role of the indicated proteins on tumorigenesis.
Nevertheless, in my opinion, the text and table 1 should be revised. In table 1
Author Response
We thank the reviewer for critical review of our manuscript. We incorporated changes in the manuscript as suggested by reviewers. The point wise answers to reviewers comments are as below:
Comment 1) Please check and uniform all the abbreviation in the text: page 1 line 41 RB is abbreviate but not in the rest of the paragraph. The same issue in page 6 line 223 and 233. Page 8 lane 269 uveal melanoma is abbreviate as UM, but not mentioned for the first time.
Answer: The abbreviations are modified throughout the manuscript as suggested by the reviewer.
Comment 2) The figure 1 is not consistent with the sentence reported at page 4, lanes 148 to 154. Perhaps the authors refer to another figure. Please check
Answer: As rightly suggested by the reviewer, the modification is done in the revised manuscript (Lines 77,78, 91,156)
Comment 3) Table 1 is unclear. We recommend to add information on the role of the indicated proteins on tumorigenesis. Nevertheless, in my opinion, the text and table 1 should be revised.
Answer: We agree with the reviewer. The miRNA in the RB serum and RB tumor were either down regulated or upregulated compared to healthy controls, Unlike in RB cells, the miRNA which is detected in the exosomes is listed. The title of the column was corrected to avoid confusion. In the manuscript, we want to highlight that miR 9 and miR 494 has maximum number of targets in the proteins detected in the retinoblastoma exosomes. Among the target proteins, 23 are targets for both miR 9 and miR 494. We have discussed about all the relevant proteins and their role in tumorigenesis in the revised manuscript (lines 362-401).
Reviewer 2 Report
The review article „Exosomes: Insights from Retinoblastoma and other Eye cancers” summarize the research in the field of eye exosomes, with emphasis on retinoblastoma.
Sections 1 and 2 (Introduction and Exosomes of the eye) are well written providing a good overview of the current research.
Part 3 “Eye cancers”: The authors summarize facts of different eye cancers without any information about exosomes. Only the last sentence is a general statement about exosomes without relationship to the eye cancers discussed before. If the authors would like to use the title „Exosomes: Insights from Retinoblastoma and other Eye cancers” they have to give information about exosomes in “other Eye cancers” and not only for Retinoblastoma.
Line 154: I cannot see the connection between the sentence and Figure 1?
Line 230-246: The authors wrote a large part about miRNAs in section 4 while the miRNA section is actually the next section. For me it does not make sense in this context.
Section 5 “miRNA an their targets in Retinoblastoma”: This is the main part of the manuscript. The authors are very much focusing on miRNAs and their targets in the context of exosomes in RB. Table 1 summarize miRNAs and their target and I was wondering if there is no information about downregulated miRNAs in RB cell lines exosomes? In Table 2 common targets of two miRNAs are listed and the authors described some of them [10 out of 23] in detail in the manuscript. However, it is not obvious why the authors select those 10 and not the others?
Author Response
We thank the reviewer for critical review of our manuscript. We incorporated changes in the manuscript as suggested by reviewers. The point wise answers to reviewers comments are as below:
Comment 1: Part 3 “Eye cancers”: The authors summarize facts of different eye cancers without any information about exosomes. Only the last sentence is a general statement about exosomes without relationship to the eye cancers discussed before. If the authors would like to use the title „Exosomes: Insights from Retinoblastoma and other Eye cancers” they have to give information about exosomes in “other Eye cancers” and not only for Retinoblastoma.
Answer: The available information on exosomes in other eye cancers is included in the revised manuscript (lines:180-186; lines: 190-194; Lines: 198-201)
Comment2: Line 154: I cannot see the connection between the sentence and Figure 1?
Answer: As rightly suggested by the reviewer, the modification is done in the revised manuscript (Lines 77,78, 91,156)
Comment 3: Line 230-246: The authors wrote a large part about miRNAs in section 4 while the miRNA section is actually the next section. For me it does not make sense in this context.
Answer: The miRNA in section 4 is shifted to the section 5 (miRNA and their targets in Retinoblastoma)
Comment 4: Section 5 “miRNA an their targets in Retinoblastoma”: This is the main part of the manuscript. The authors are very much focusing on miRNAs and their targets in the context of exosomes in RB. Table 1 summarize miRNAs and their target and I was wondering if there is no information about downregulated miRNAs in RB cell lines exosomes? In Table 2 common targets of two miRNAs are listed and the authors described some of them [10 out of 23] in detail in the manuscript. However, it is not obvious why the authors select those 10 and not the others?
Answer: The point raised by the reviewer is well taken and addressed in the manuscript. The miRNA in the RB serum and RB tumor were either down regulated or upregulated compared to healthy controls, Unlike in RB cells, the miRNA which is detected in the exosomes is listed. The title of the column was corrected to avoid confusion.
Among the target proteins, 23 are targets for both miR 9 and miR 494. In the revised manuscript all the relevant proteins and their role in tumorigenesis is discussed(lines 362-401)